# Entropy and Information within Intrinsically Disordered Protein Regions

**DOI:** 10.3390/e21070662

**Published:** 2019-07-06

**Authors:** Iva Pritišanac, Robert M. Vernon, Alan M. Moses, Julie D. Forman Kay

**Affiliations:** 1Program in Molecular Medicine, The Hospital for Sick Children, Toronto, ON M5G 0A4, Canada; 2Department of Cell & Systems Biology, University of Toronto, Toronto, ON M5S 3B2, Canada; 3Department of Computer Science, University of Toronto, Toronto, ON M5T 3A1, Canada; 4Centre for the Analysis of Genome Evolution and Function, University of Toronto, Toronto, ON M5S 3B2, Canada; 5Department of Biochemistry, University of Toronto, Toronto, ON M5S 1A8, Canada

**Keywords:** intrinsically disordered proteins, Shannon entropy, information theory, evolutionary conservation, conformational entropy, low-complexity sequences, liquid-liquid phase separation, post-translational modifications, conformational ensembles, biophysics

## Abstract

Bioinformatics and biophysical studies of intrinsically disordered proteins and regions (IDRs) note the high entropy at individual sequence positions and in conformations sampled in solution. This prevents application of the canonical sequence-structure-function paradigm to IDRs and motivates the development of new methods to extract information from IDR sequences. We argue that the information in IDR sequences cannot be fully revealed through positional conservation, which largely measures stable structural contacts and interaction motifs. Instead, considerations of evolutionary conservation of molecular features can reveal the full extent of information in IDRs. Experimental quantification of the large conformational entropy of IDRs is challenging but can be approximated through the extent of conformational sampling measured by a combination of NMR spectroscopy and lower-resolution structural biology techniques, which can be further interpreted with simulations. Conformational entropy and other biophysical features can be modulated by post-translational modifications that provide functional advantages to IDRs by tuning their energy landscapes and enabling a variety of functional interactions and modes of regulation. The diverse mosaic of functional states of IDRs and their conformational features within complexes demands novel metrics of information, which will reflect the complicated sequence-conformational ensemble-function relationship of IDRs.

## 1. Information—Central to the Central Dogma

The concept of information is deeply rooted in the central dogma of molecular biology. According to the simplest version of the dogma, a coding stretch of DNA encodes the information for RNA, which in turn encodes a protein that encodes a function [1]. When defining the dogma, Crick specified that information refers to “the precise determination of sequence, either of bases in the nucleic acid or of amino acid residues in the protein.” [1]. Given that the essential molecules of life act as encoders and transmitters of information, life itself can be considered a manifestation of the flow of information (Box 1). Indeed, the ability to process information has been proposed as one of the criteria to define life [2]. Since the discovery that the 3D structures of folded proteins are encoded in their amino acid sequence it has been understood that this genetic information translates to structure and function through a thermodynamic process that minimizes free energy, which for structured proteins involves forming functional ordered conformations [3]. This structure-function relationship makes the determination of 3D structures of proteins a major focus of biological research [4] with structural information used to rationalize molecular mechanisms and therapeutic design [5].

### 1.1. Information in IDRs—Problems for The Paradigm

Proteins that lack stable tertiary structure in their native form, known as intrinsically disordered proteins and regions (IDRs), comprise a large fraction of the eukaryotic proteome [6,7]. Many studies indicate that IDRs often exhibit large evolutionary sequence variation [8,9,10] and sample a vast 3D conformational space [11]. However, the apparent randomness of IDR evolution and 3D conformations is at odds with the diversity and significance of IDRs’ biological roles in regulatory [8,12,13] and signaling processes [14,15], and their frequent implications in disease [14,16,17,18]. Recent methodological developments are increasingly unveiling “hidden” information in IDRs and in so doing are forcing us to reconsider how thermodynamic protein behavior can translate genetic information to function. Here, we discuss entropy in the sequences and conformational ensembles of IDRs and the requirement for new ways of converting entropy to information in the context of their biological functions. 

We briefly outline the concepts that are essential for understanding the terminology (Section 1). In Section 2, we argue that methods based on positional information in multiple sequence alignments need to be abandoned in order to quantify the information in the sequences of IDRs. In Section 3, we discuss experimental techniques that characterize various aspects of the high conformational entropy of IDRs. We explore how conformational plasticity is utilized for the interactions of IDRs with their targets, and how it is impacted by post-translational modification and liquid-liquid phase separation. We conclude by discussing future perspectives for defining and exploiting the information that is available in the sequences and conformational ensembles of IDRs (Section 4).

### 1.2. Uniting Different Entropies and Extracting Information 

Entropy is a concept that transcends and unites different areas of science [19,20]. The term itself was born with thermodynamics [21], with subsequent atomic interpretation of entropy giving foundations to statistical mechanics [22,23]. In the general terms of information theory, entropy is a measure of the uncertainty in the identity of a random variable or the state of a system, given as:(1)$$H\left(P\right)=-K\underset{i}{\sum }p_{i}log_{2}p_{i}$$
where *K* is a positive constant and *P* stands for the probability mass function (PMF), $P=\{p_{1}\ldots p_{n}\}$, or the probability of a random discrete variable or a system to be found in states *i* = {1,2…*n*}. If the logarithm in Equation (1) is base 2, entropy is measured in bits. In information theory, *K* is set to unity, whereas in statistical mechanics, *K* becomes the Boltzmann constant (*K* = *k_B_* = 1.3808 × 10^−23^ J/K). In his pivotal work, Shannon [24] demonstrated that Equation (1) captures the intuitive notions of entropy: it is positive, increases with increasing uncertainty, and is additive with respect to the independent sources of uncertainty.

In practice, to measure information content, relative entropy or Kullback-Leibler (KL) divergence is often employed, which measures entropy of a probability distribution with respect to another probability distribution, given as:
(2)$$D\left(P∥Q\right)=\underset{i\in A}{\sum }p_{i}log_{2}\frac{p_{i}}{q_{i}}$$
where *P* and *Q* are defined over a finite set *A*, *i*
∈
*A*. KL divergence can be computed between a reference probability distribution (*Q* = *P**), and the measured one (P) to evaluate the dissimilarity between the two. Here we refer to the KL divergence between the distribution of symbols in biological sequences under “total uncertainty” (*Q*) and the observed distribution (*P*) as the “Sequence Information”, which is distinct from the “Mutual Information,” (Box 2) a more formal measure of information that is also widely used in biological sequence analysis.

## 2. Sequence Entropy Metrics Fail to Extract Information from IDRs

### 2.1. Sequence Entropy Can be Computed Horizontally and Vertically

Complexity, information, and entropy have been difficult to apply for practical gain in many biological settings. However, Shannon entropy and related information theoretic measures, such as relative entropy and mutual information (Box 2), have been widely applied to study biological sequences [25]. To apply these concepts, biological sequences are thought of as strings of symbols, the four “letters” of DNA and RNA and the 20 of proteins. If these letters are interpreted as messages from an information source in the Shannon sense, the entropy in a biological sequence can be computed using the standard formula (Equation (1)).

Box 1.What is information?Information is the reduction in entropy (here thought of as uncertainty) in the receiver [26]. In the biological context of the central dogma, we can imagine an mRNA carrying information to the ribosome. Before the mRNA arrived, the ribosome might be expected to assemble polypeptides with amino acids in a random order. However, once the mRNA arrives, out of the astronomically many sequences that are possible, the ribosome assembles one specific polypeptide sequence (note that this is more than one in reality because of noise, i.e., errors in the translation process). Hence, the entropy of the produced polypeptide is dramatically reduced: information has been transmitted. This example is instructive because it highlights the dependence on the receiver, in this case the ribosome: if the ribosome did not recognize the mRNA as a signal, no information would be transmitted.Thus, information is measured as the difference in entropy before and after a probabilistic event, a measurement, or receiving of a message:
(3)$$I\left(X\right)=H_{before}-H_{after}$$The maximum information is gained when *H_after_* equals 0 and thus all *H_before_* is converted to information, so the entropy sets the maximum for the possible information transmitted. This, however, need not be the case in practice, as communication channels can contain noise or some degree of uncertainty can remain after receiving a message or performing a measurement (*H_after_*> 0). Note that relative entropy and information are often used interchangeably in the bioinformatics literature.

Shannon entropy can be measured horizontally, i.e., in a window across a protein sequence to detect a statistical bias in the use of the 20 amino acid alphabet in protein sequences [27,28,29]. In practice, the bias is evaluated relative to a uniform or an empirical background distribution (Equation (2)). The sequences with reduced amino acid alphabets will show reduced (relative) entropy values, which can be used to classify protein sequences as ‘high’ or ‘low’ complexity, with some IDRs sequences displaying low complexity [27,28,29] (see Section 2.5). There are statistical limitations of such horizontal entropy evaluations due to limited samples, and it is unclear whether and how these metrics on their own could be used to extract information specific to a protein function. An alternative way to measure entropy as a way to gain functionally relevant information from biological sequences is to focus on a functionally equivalent set of sequences and evaluate entropy vertically across their alignment [30,31].

Box 2.Mutual information.Mutual information is another entropy-based metric often employed to measure the (in)dependence of two random variables or probability distributions. If two random variables *X* and *Y* are independent, then their joint probability distribution, *P*(*X*,*Y)*, will be equal to the product of their individual probability distributions:
(4)P(X,Y)=P(X)P(Y)The degree of dependence can be revealed by computing the relative entropy (KL divergence, Equation (2)) between the joint probability distribution and the product of the individual distributions:
(5)$$MI\left(X;Y\right)=D\left(P\left(X,Y\right)∥P\left(X\right)P\left(Y\right)\right)=\underset{i,j}{\sum }P\left(x_{i},y_{j}\right)\log_{2}\frac{P\left(x_{i},y_{j}\right)}{P\left(x_{i}\right)P\left(y_{j}\right)}$$For independent variables for which Equation (4) holds, the mutual information is equal to zero. Otherwise, the mutual information corresponds to a reduction in entropy, i.e., a gain in information, about the outcome of *X* given the knowledge of *Y.* The latter is best expressed in an equivalent, theoretical expression of mutual information that uses the concept of conditional entropy, which corresponds to the entropy of *X* given the knowledge of *Y*, *H(X|Y)*, and vice versa *H(Y|X)*:
(6)MI(X;Y)=H(X)−H(X|Y)=H(Y)−H(Y|X)In studies of biological sequences, mutual information is computed between the columns of a multiple sequence alignment to evaluate if neighboring, or distant, positions along the sequence are independent or covary. To this end, the frequencies of co-occurrence of elements at potentially covarying positions are used for $P\left(x_{i},y_{j}\right)$ in Equation (5) [32]. In other applications, mutual information is often difficult to use because the joint distribution (in Equation (5)) between the biological receiver and signals is difficult to obtain. For example, to compute mutual information between a protein binding event and a peptide sequence, even under the assumption that there are only two states (bound and unbound), the joint distribution includes all the unbound peptide sequences, which are, in practice, rarely known. In contrast, the KL divergence (Equation (2)) can be computed for the distributions between uncertain (before) and bound (after) states, which does not require the full joint distribution.

### 2.2. Sequence Entropy in Biological Macromolecules: The “Positional Information Paradigm”

In the ‘vertical’ applications, each position in a set of aligned biological sequences is regarded as an independent message, and relative entropy is therefore calculated at each position. The probability, *P*, is simply a multinomial distribution on the observed numbers of each type of symbol at that position. The relative entropy at each position can be computed with respect to the “maximum” entropy for that sequence type: log_2_(4) for DNA and log_2_(20) for proteins, or alternatively, relative to a defined background distribution (*Q* in Equation (2)). This can be, for instance, the overall distribution of nucleotides or amino acids in the aligned sequences [32]. The relative entropy at each position thus corresponds to information and forms the basis for the widely used “sequence logo” representation for biological sequences [33] (Figure 1A). In this representation, the heights of the letters in this representation are proportional to the information at each position [33]. Since information is additive, to obtain the information in the whole alignment, the information at each position is simply added.

The information at each position in alignments has been found empirically and theoretically to be predictive of biological function, where information is typically associated with the concept of sequence conservation. In classical studies, information at each position in DNA sequences patterns has been found to predict protein binding in vitro and in vivo [35,36]. Similar results have been found for the information in protein sequences: positions with large amounts of information point to the functional residues [37,38]. In fact, protein folding has been proposed as ‘a noiseless communication channel’ [39] leading from sequence to structure, whereby all information in the protein structure is shared with the sequence (mutual information) [30,39]. The positional information and the correlations (mutual information) between positions are being used in predictions of protein structure [40,41,42,43], catalytic residues [44] and protein interaction interfaces [45].

### 2.3. Evolutionary Origin of Positional Information in Sequence Alignments

Biological sequences represent the products of the evolutionary process. Sequences are the “genotypes” that produce biological functions and phenotypes. Hence, if natural selection is acting to preserve a biological function, changes (mutations) in the sequence (genotype) that alter that function will be removed from the population over evolutionary time. It has been recognized since the early days of molecular evolution that natural selection would act to reduce the entropy and increase the information in the genotype [46]. Indeed, modern bioinformatics analyses have borne out this prediction: sequence positions with more information show less evolutionary variation [47] and stronger natural selection [48].

Thus, theoretical, empirical, and evolutionary arguments all support the application of information theory to the positions in alignments [30,31,49], and the use of information as a measure of the biological importance of that position. So appealing is this “positional information paradigm” that the arguments have been presented for a 1:1 relationship between sequence information and function [49]: if there were no detectable information at some positions of a sequence alignment, this could rule out the possibility of function for those positions. In other words, positions in sequences with no information have been considered to encode no function.

### 2.4. Intrinsically Disordered Regions Contain Little Positional Information, but Still Encode Function

Most positions in disordered regions are not readily aligned across long evolutionary distances [50]. When homology can be inferred based on surrounding folded domains, many of the positions in disordered regions show nearly maximum sequence entropy, as expected for random sequences (Figure 1). IDRs are known to contain molecular recognition features (MoRFs) that are characterized by disorder-to-order transition upon interactions, and short-linear peptide motifs (SLiMS) that mediate protein-protein interactions and are often associated with posttranslational modifications [8,51,52,53,54]. In some cases, these functional regions of IDRs show clear positional information [54]. However, the information is limited to short stretches of amino acid residues, typically three to fifteen amino acids long [55]. Hence, alignments of IDRs reveal little sequence information, as measured by the definitions introduced above, and the functional importance of most amino acid residues in disordered regions remains unknown. Thus, application of the positional information paradigm is consistent with the idea that most IDRs encode no biological function (“junk proteins” [56]) or serve as inert linkers or tethers that harbor interaction sites (MoRFs or SLiMs) needed to bring other components together [57].

Nonetheless, there is increasing appreciation of the critical biological function of disordered regions with little positional information [9,10]. How can we reconcile this contradiction? We argue that the positional approach to information primarily measures the spatial relationship between residues, i.e., functions specific to the exact position in a folded structure that a residue occupies. If a protein encodes a functional structure, then the metric coincides with function. In fact, residues in folded proteins evolve according to their spatial position in a protein structure. No two residues can occupy the same position in space, and when a position in a structure has a unique functional role no two residues can share that function. This uniqueness of positional information in folded proteins underlies most methods involved in positional analysis of sequence alignments (see Section 2.2). In contrast, if a protein functions as an ensemble of conformers (Section 3), where multiple residues can occupy the same functional positions in space dynamically over time, then the information on function remains undetectable when attempted to be revealed through a multiple sequence alignment. This consideration illustrates how the relationship between space and function in IDRs is significantly more complicated than the 1:1 relationship characteristic of folded proteins.

Given that most of an IDR amino acid sequence does not need to encode stable residue-specific interactions, what kind of, and how much information do IDRs contain? Studies of the sequence determinants of IDR function for individual proteins have revealed “cryptic” sequence features that are correlated with specific molecular functions [58]. This suggests that IDR sequences do contain some form of information, but that it is not detectible using sequence alignment-based techniques, which assume that each position carries information independently. Recently, we and others have begun to detect evolutionary signatures in disordered regions that are largely devoid of positional information [9,10,59,60]. This revealed that most of the apparently highly diverged IDRs in the yeast proteome contain multiple conserved molecular features in the form of physicochemical properties (e.g., net charge, isoelectric point, hydrophobicity, fraction of polar residues, etc.), amino acid repeats (e.g., RG, RGG, QQ), sequence complexity, and sequence motifs (e.g., phosphorylation motifs, proline-rich motifs, nuclear localization signals, mitochondrial localization motifs, etc.) [10]. The evolutionary signatures in IDRs establish how natural selection can preserve distinct molecular features predictive of function, and suggest that, in addition to sequence motifs, bulk properties of disordered regions encode functional information stored in IDRs [10].

A number of functional roles of IDRs have been identified that do not rely on the formation of stable, folded structure: proteinaceous detergents and solubility tags, scaffolding within interaction hubs, entropic springs, timers and linkers, allosteric transmission, and binding via bulk electrostatic or other physical property. Other important functions of IDRs that do not generally require stable structure are in cellular compartmentalization and biomaterial formation through the process of liquid-liquid phase separation, with consequences for regulation of enzymatic reactions and other biological processes [61,62,63,64,65,66,67,68]. The lack of stable structure exposes IDRs for interactions with multiple binding partners, which can enable more efficient driving of macromolecular condensation. In addition to the dynamic multivalent interactions of sequence motifs within IDRs with modular binding domains of folded proteins, IDRs also contribute to phase-separation through their low-complexity (LC) regions, which can participate in multi-valent protein and nucleic acid binding [69,70,71,72,73,74]. A number of studies have assessed the extent to which conformational dynamics and secondary structural propensities of IDRs are affected by phase-separation [69,73,75,76,77]. This revealed that IDRs can remain highly disordered in the phase-separated droplets [68,69,70,75], while exhibiting significantly slower translational diffusion [69,75]. Inter-molecular contacts within the droplets could be detected [69,75,76], with an increase in intra-molecular contacts also reported in some cases [76]. It is interesting to note that sequence alterations and disease-associated mutations in IDRs can directly impact phase separation propensities [69,78,79]. For instance, disease-associated mutations in hnRNPA2 and FUS were found to promote aggregation [72,78,79], while some ALS-associated variants of TDP-43-LC disrupt phase separation by destabilizing the transient formation of a helical conformation [71].

### 2.5. Information in Low-Complexity IDR Sequences

As defined above, sequence entropy assumes that each symbol in a protein, where symbols correspond to 20 amino acids, carries independent information. In most proteins this assumption holds, as individual amino acids make stable inter- and intra-molecular contacts and the observed distribution of amino acids is the result of selection acting on random point mutations in the DNA sequences that encode them. Correlations between positions will reduce positional entropies due to functional associations between positions [30,31]. However, when DNA and protein sequences are the result of replication machinery errors (e.g., unequal cross-over, replication ‘slippage’) [80,81] each symbol no longer has the full range of possibilities, i.e., the next amino acid in the sequence can only be one of the previous ones that has been copied and inserted. These mechanisms have generated a large class of sequences in the genome known as “low-complexity”, which show a reduced amino acid (or DNA) alphabet. These sequences are typically removed from standard bioinformatics analyses because they violate many of the assumptions of the methods [28,82,83]. Paradoxically, low-complexity, by definition, implies low entropy and therefore, when compared to the entropy of complete uncertainty, large information, which in this case is very different from that in globular folded protein domain sequences. We note that this “information” is likely to be a result of the different mutation processes that generate these sequences [56,80,81], and it is unclear in many cases what is, or whether there is, a “receiver” for this “information.”

Use of “horizontal” sequence entropy metrics (across a protein sequence) revealed that IDRs often exhibit low-complexity, however, they are not limited to it [29]. Similarly, low-complexity sequences are not always disordered and detection of low-complexity alone is not predictive of any particular protein sequence property or function [66]. Nonetheless, categorization of low-complexity IDRs based on physicochemical characteristics can be used to classify these sequences based on their properties [66]. In this context, there is a considerable association of low-complexity IDR sequences with phase separation mediated through π-π interactions [84] or backbone beta interactions that can also lead to fiber formation [85,86]. Figure 2 shows that while low-complexity human protein regions are more often predicted to phase separate there is additional information content in factors such as specific composition and length, with even the total sum of glycine residues containing information not captured by Shannon entropy. Further developments in this direction are important given the enrichment of low-complexity IDRs in biomolecular condensates [87,88,89].

Similarly, functionally relevant insight can be gained from correlations between features of the primary amino acid sequence and the observed phase separation behavior of IDRs in vitro and in cells [70,84,90,91,92,93]. In addition, polymer physics models are being built to understand the competing enthalpic and entropic contributions to the free-energy changes that underlie the phase separation of IDRs, and the dependence of these contributions on the primary amino acid sequences and post-translational modifications (PTMs) [66,94,95,96]. Ongoing research in these areas, jointly with insights from cell biology studies, will further clarify how information stored in IDR sequences encodes their phase-separation behavior, and how this is tuned by the changes in the sequence and/or environment. 

## 3. IDRs Feature High Conformational Entropy

The high positional (“vertical”) entropy measured from multiple sequence alignments of IDRs (Figure 1) stems from their lack of stable three-dimensional structures, which in turn underlies their high conformational entropy [6,97,98,99,100] (Figure 3). The high conformational entropy of IDRs has been proposed as a thermodynamic reservoir of free energy (Box 3) that can be used to regulate their functions and interactions [15,101,102,103,104].

Box 3.Entropy in biophysics.Although equilibrium thermodynamics and information theory share the mathematical expression for entropy (Equation (1)), the term is used in different contexts in the studies of proteins in bioinformatics (see above) and biophysics. In biophysics, entropy is defined, calculated, and measured on multiple levels. The total number of possible arrangements, i.e., different relative center-of-mass positions, of polypeptide chains in solution contributes to configurational entropy [95,109]. Configurational entropy encompasses conformational entropy, which is defined by the number of different conformations of an individual polypeptide chain. Conformational entropy itself can be split into two contributions: local, small-scale fluctuations about a well-defined structure (e.g., an α-helix), and larger-scale conformational differences (e.g., different protein conformations in a random coil ensemble) [110,111]. In an experimental system, the free energy change accompanying protein folding or an interaction with a protein or ligand is routinely measured using, e.g., isothermal titration calorimetry [112]. This measurement allows decomposition of the free energy change, $\Delta G_{tot}=\Delta H_{tot}-T\Delta S_{tot}$, into the total enthalpic (Δ*H_tot_*) and entropic (−*T*Δ*S_tot_*) contributions. The entropic component encompasses contributions from all sources of entropy in the system, including the conformational entropy of the protein and its interacting partner (e.g., ligand), the contributions from solvent and the roto-translational degrees of freedom of all interacting partners, and any other, remaining contributions; $\Delta S_{tot}=\Delta S_{protein}^{conf}+\Delta S_{ligand}^{conf}+\Delta S_{solvent}+\Delta S_{r-t}+\Delta S_{other}$ [113]. Untangling the different entropic contributions to the free-energy change, however, remains non-trivial.

### 3.1. Conformational Entropy of IDRs is Difficult to Measure Precisely

In contrast to folded proteins that show functionally important conformational dynamics about a well-defined, energetically stable structure, IDRs display large conformational heterogeneity [8,114]. The ease of interconversion between different conformational states of IDRs is well captured by a rugged, relatively flat energy-landscape model with abundant minima separated by small energy barriers [8,101]. Thermodynamically, the structural heterogeneity of IDRs stems from the large contribution of conformational entropy to their free energy (G = H − TS). This entropy contribution becomes a critical component to the free energy change between the different states of IDRs, for instance, upon post-translational modification or complexation with a target biomolecule (Box 3). 

Despite our abstract understanding of the conformational entropy as a defining characteristic of IDRs [6,97,98,99,100], it has proven tremendously difficult to quantify the full range of this thermodynamic component at IDRs’ disposal. Some of the underlying reasons are the limited availability of experimental data to characterize the vast degrees of freedom that contribute to the conformational entropy of IDRs, the lack of understanding of the contributions of solvent, and a still evolving synergy between theory, experiment and simulations (Box 4). Therefore, our understanding of conformational entropy, and its change in a functional context (Figure 4), is limited to the degrees of freedom that can be measured experimentally; increasing contributions from a range of experimental techniques of varying resolutions are helping gain a more conclusive picture [11,115,116,117]. In addition, molecular dynamics simulations can often enhance the interpretability of experimental parameters [118,119,120] (Box 4).

Box 4.Conformational entropy of IDRs.Solution-state NMR spectroscopy is the primary experimental technique for the dynamical characterization of proteins at atomic resolution [121,122,123,124,125], which can serve as a proxy for conformational entropy. In the simplest terms, the conformational entropy of a protein can be described by the total number of conformations that are accessible to the protein under a defined set of macroscopic conditions (e.g., protein concentration, temperature, and volume). The full extent of the conformational space of a protein, however, is prohibitively large. Therefore, to compute conformational entropy approximately, conformations are discretized and defined using a combination of structural parameters. In experimental practice, only sparse and typically strictly local structural parameters can be measured for disordered proteins, e.g., distributions of the backbone Φ and Ψ angles (Figure 3A). These can be used to constrain molecular dynamics (MD) or Monte Carlo (MC) simulations and estimate the conformational entropy of a protein.To understand the restriction of conformational entropy in a disordered polypeptide chain, both short- and long-range restraints are essential. Therefore, continuous efforts are being put forth community-wide [126] to improve the measurement of long-range restraints, correct for any noisy contributions to the restraints (e.g., the error in back-calculations of experimental observables from structures), and integrate information available from complementary, lower-resolution techniques such as small-angle X-ray scattering (SAXS) and single-molecule fluorescence (SMF) [115,116]. These efforts are aided by the continuously improving databases of reference random coil chemical shifts [127] and predictions of NMR observables from statistical coil models [11]. To generate adequate IDR ensemble representations, programs like ENSEMBLE [128] and ASTEROIDS [129] use statistical coil, structurally biased or MD-derived models to initially generate a large pool of conformers that are then sub-sampled to optimize the agreement with a combination of complementary local and global experimental restraints (including NMR chemical shifts, residual dipolar couplings, J-couplings, paramagnetic relaxation enhancements, and nuclear Overhauser effects, SAXS and SMF data). In parallel, a better understanding of the dynamics of inter-conversion between IDR conformers in free and bound-states using spin relaxation and chemical exchange NMR techniques is being sought [11,122,130,131]. Additional insights can be provided by molecular dynamics simulations restrained by experimentally available information [119,120,132]. Further technological advances in NMR spectroscopy in synergy with lower-resolution techniques and improved computational approaches [126] are expected to bring us closer to understanding conformational entropy of IDRs in isolation, and its changes upon PTMs [133,134], interactions with ligands [135,136], and formation of complexes [103].

The extent of backbone conformational sampling of an IDR in Ramachandran space can be obtained with reasonable precision from the local structural parameters [105] (Figure 3A); however, experimental information is much more sparse for correlations between the backbone angles of distant sites along the chain. Consequently, reconstructing the extent of the conformational sampling, i.e., conformational entropy, of an entire IDR quickly becomes a severely underdetermined problem that grows exponentially with the size of the IDR [105,106].

### 3.2. Conformational Diversity and Information in Ensembles of IDRs

The degree of backbone flexibility, local secondary structure propensities, distributions of short- and long-range contacts, and the overall shape and size of IDR ensembles represent experimentally obtained information from IDRs in solution [11,115,116,141,142] (Box 4). On these bases, a picture emerges of the highly diverse conformational features of IDRs in their isolated states (Figure 3B). While some IDRs locally resemble a random-coil [106], but show some compaction due to transient long-range contacts [143,144], others are more compact and appreciably sample transient secondary structure in their free form [139].

The sampling of local transient secondary and tertiary structure in the free forms of IDRs (Figure 3B) is informative, as it can be predictive of their interactions with ligands [145,146] and protein binding partners [147,148,149], and has been found to impact the kinetics and lifetimes of these interactions [150,151]. For instance, differential propensity to form an α-helical conformation in the free state of disordered regions of two transcription factors (TFs) that bind to the same site on a transcriptional coactivator, was shown to result in different binding mechanisms [150]. The TF that rapidly sampled an α-helical conformation in its free state was shown to interact with the transcriptional activator through conformational-selection. In contrast, the second TF interacted with the transcriptional activator by an induced-fit mechanism [152]. These in vitro observations were proposed to directly relate to the biological functions of the two TFs, as one represents a constitutive transcriptional activator, whereas the other requires phosphorylation for high-affinity binding to the transcriptional coactivator [150]. Another example demonstrated the impact of a transient, partially helical conformation within the disordered N-terminal transactivation domain (TAD) of p53 [148]. The engineered increase in the transient occupancy of the helical conformation in p53-TAD enhanced the binding affinity for its cellular binding partner and regulator E3 ubiquitin ligase Mdm2, which resulted in disrupted p53 signaling activity and downstream gene regulation in a cellular context [148]. Hence, amino acid substitutions in IDRs can affect transient, local structural propensities [148], and can also exhibit longer-range effects through the disruption of stabilizing tertiary interactions between distant secondary structural elements [131].

In conclusion, IDRs retain a high amount of conformational entropy in their native states, unlike the low entropy states occupied by stable folded proteins. In some cases, IDRs may have conformational entropy that is comparable in magnitude to that of an unfolded polypeptide chain. This conformational plasticity could aid IDRs in sampling a diverse range of distinct functional states or regulate the binding behavior of a single functional conformation. We used the examples above to illustrate how the formation of transient structure in IDRs can, in some cases, reduce the uncertainty about the functional conformational state(s), thereby comprising functionally relevant structural information. Such findings motivate the implicit search for, and utilization of, mutual information between the sequences of IDRs and their structural propensities [153]. Nonetheless, the absence of measurable structural propensities in the free states of IDRs, or in their bound states (see below), does not equal functional irrelevance.

### 3.3. IDRs Can Retain High Conformational Entropy in Complexes

The absence of transient secondary or tertiary structure in the free state of an IDR does not always equate to a complete absence of structural propensities in a complex (Figure 4A). IDRs engage in diverse complexes [154] featuring complete or partial folding upon binding [131,152,155,156], remaining highly disordered as in “extreme” disordered discrete complexes [157] or phase-separated states [68,69,70,75], exchange between multiple highly disordered states [138,157], or retention of some secondary and tertiary structural propensity while remaining disordered in the non-interacting regions [103] (Figure 4A). Reports of mixed ordering in some parts of an IDR with increased disorder in other segments are particularly interesting, as they suggest fine-tuning of the entropic loss upon complexation [101,139,158,159], while maintaining a biologically meaningful binding affinity [159]. It is also interesting to note that SLiMs in IDRs need not always rigidify upon binding, as might be thought due to the bias in X-ray structures. In contrast, SLiMs can exhibit fast dynamics on the surface of a binding partner [158], or dynamics on the intermediate timescale in the context of exchange with other binding elements within dynamic, multivalent complexes [139].

The parameters obtained from NMR dynamics measurements of fast (ps-ns) backbone and side-chain dynamics have been introduced as a proxy for contributions to changes in conformational entropy upon protein folding [160], and interactions of folded proteins with their binding partners [113,161,162] (Box 4). An approach for addressing conformational entropy changes in both IDRs and their binding partners upon interaction was proposed on these bases [159]. However, IDRs can often have slower conformational exchange on the NMR chemical shift timescale, e.g., cis-trans Pro isomerization leading to distinct resonances [163], and a range of stabilities of intramolecular interactions leading to some exchange-broadened resonances [164]. In addition, IDRs can exhibit slower conformational exchange in the context of dynamic complexes, which leads to linewidth broadening [165,166,167] and, while useful in some cases, can hinder attempts to obtain information on the conformational sampling in these protein regions. These features of IDRs challenge the assumption that their conformational entropy can be estimated from fast timescale motion alone. Finally, as IDRs engage in extensive protein-solvent interactions, a better understanding of the solvent contributions to the changes in entropy upon complexation and phase separation involving IDRs (see below) will be critical to fully understand the driving forces behind their biomolecular interactions.

### 3.4. Post-Translationally Modified Sites in IDRs Transmit Biological Information 

The conformational plasticity of IDRs is also exploited for the regulation of their biological activities through post-translational modifications (PTMs) [8,102,134,168] (Figure 4). IDRs are the prevalent sites of PTMs perhaps because the lack of stable structure enables easier access to modifying enzymes, as previously proposed [134,169,170,171,172]. The PTM sites represent high-information density regions in IDR sequences that offer vastly diverse options for regulating biological functions, such as modulation of subcellular localization, protein-protein interactions, and rates of protein-synthesis [134,169,170,171,172].

From the structural point of view, PTMs can act on a broad scale, from modulating secondary structural propensities [140,173] (Figure 4Ci) and transient tertiary intramolecular contacts, to completely changing the fold of a protein [133,134] (Figure 4Cii). For instance, N-terminal acetylation of α-synuclein was shown to increase the population of a transiently formed α-helix at the N-terminus and subsequently increase the binding affinity for lipids [140,173,174] (Figure 4Ci). Importantly, this result has been confirmed in living cells by NMR spectroscopy, demonstrating how a PTM can establish a structure-function link in an IDR [175,176]. A more drastic structural impact of PTM was shown on 4EBP2: upon multisite phosphorylation, a 40-residue region of the disordered protein transitioned to a folded structure (Figure 4Cii). This structural conversion leads to the sequestration of its eIF4E binding interface, thus providing mechanistic insight into its control of translational initiation [133].

In addition to their effect on specific interactions of IDRs with other cellular components, PTMs can also regulate liquid-liquid phase separation [68,70,76,77] (Figure 4D). For instance, phosphorylation was found to promote phase separation of some IDRs, such as Tau and FMRP [68,76], while inhibiting that of others, e.g., FUS-LC [77]. Similarly, arginine methylation was found to inhibit phase separation in some contexts [68,70,72], yet promote granule formation in others [177]. The modulation of phase separation by PTMs suggest a mechanism to control both the formation and dissolution of different membranelles organelles [68,70].

Post-translationally modified sequence motifs in IDRs are sometimes positionally conserved and can manifest as information due to the reduction in the positional sequence entropy. In other instances, PTM sites are not conserved at precise positions along the sequence, but as an aggregate property [178]. This illustrates how functional information in IDRs can be encoded in the absence of positional sequence conservation [9,10], as discussed in Section 2.4.

### 3.5. Functional Engagements of IDRs Alter Physical Entropy in All Directions

The favorability of functional engagements of IDRs with other cellular components is dictated not only by entropy, but also by the underlying changes in free energy. The entropic contribution to such changes can go in either direction, depending on the enthalpic components, and both increases and decreases in the overall entropy can be observed in a functional context. Hence, a decrease in physical entropy cannot be used directly to extract functionally relevant information in IDR-containing systems. When separately considering the constituents of the overall entropy, e.g., conformational entropy, it is often not straightforward to experimentally evaluate (or to simulate) its changes in a functional context. Information in the form of a loss of conformational entropy upon functional interactions is intuitively expected, and, indeed, numerous reports exist of IDRs that acquire a partial or complete fold in a complex or stabilize their pre-existing transient structural propensities.

However, there are also accumulating reports of biomolecular complexes in which IDRs remain highly disordered, or, where a loss of disorder in one part of an IDR is compensated by a gain of disorder in another (Figure 4A). Hence, functional context need not always result in a reduced conformational entropy of an IDR. Therefore, like their primary amino acid sequences, the conformational ensembles of IDRs demand additional metrics of functional information that are not strictly determined by structural propensities.

Similarly, post-translational modifications can restrict the conformational entropy of IDR ensembles, relative to their free state (Figure 4C), and the concentration of IDRs in phase-separated states can reduce the configurational entropy, which is typically compensated by a gain in entropy of the solvent (Figure 4D). However, a decrease in conformational entropy is not the rule, as there are reports of order-to-disorder transitions upon post-translational modifications [179,180] or changes in pH, temperature or redox state, which can support chaperoning activities [101,181]. Thus, when function is imposed as a condition (*Y* in Equation (6)), the physical entropy of IDR-containing systems, or its constituents, does not always decrease. The question therefore poses itself: what is the proper entropy function to measure information that is relevant to the biological functions of IDRs? From the information-theoretic perspective, the function should: (i) increase with an increase in uncertainty about the state of an IDR, (ii) be additive with respect to different sources of the uncertainty, and (iii) decrease when IDRs undergo functional engagements, thus directly translating to functionally relevant information. Given the current reports in the literature, most available entropic measures fall short on the criterion (iii), thus demanding a better understanding of the receivers of the information contained in IDR ensembles, and expansion of the existing or derivation of novel metrics to extract that information.

## 4. Conclusions and Outlook

The apparent high entropy in the sequence space of IDRs points to the ineffectiveness of the positional sequence-structure paradigm for extracting functional information stored within IDRs. To fully appreciate the extent of information encoded in IDRs, we first must bypass the conventional measures of protein fold and search for new ways of extracting information. This task remains difficult, primarily because of our incomplete understanding of the receivers of the IDR-encoded information. The receivers undoubtedly exist, as IDRs have myriad cellular roles. Nonetheless, the receivers do not have a universal mechanism for decoding information from the amino acid sequences of IDRs. While some information is obtained from positional sequence conservation, such as the position of SLiMs and some PTM sites, positionally conserved regions represent only a small fraction of an IDR sequence. Until recently, understanding of the functional contributions from the remainder of the IDR sequence was limited to individual proteins or protein classes. However, a proteome-wide approach to detect evolutionarily conserved molecular features of IDRs now enables IDR functions to be revealed directly from their sequences [10].

If the translation of genetic information to biological function is determined by thermodynamic processes for folded proteins, as codified in structure-function relationships, then it stands to reason that the thermodynamic behavior of disordered proteins also translates genetic information into other conformational ensemble-function and “transient-structure”/dynamics-function relationships. What kind of information can be extracted from IDR ensembles themselves? Measuring structural propensities in free states allows for some functionally relevant information to be extracted for a subset of IDRs from a reduction of the conformational entropy of an IDR chain. This structural information can be predictive of the binding affinities and the conformational preferences of an IDR in a complex. More interesting is perhaps the information gained as a difference between the entropy of the “off” (free) state of IDRs and the “on” (bound, post-translationally modified, or phase-separated) states. Here, a decrease in conformational entropy can accompany complex formation and PTMs, while a decrease in configurational entropy can underly phase-separation. However, the picture is complicated by alternative reports of highly disordered states of IDRs in functional complexes, and of an increase in disorder upon PTMs. Given that functional information cannot always be acquired from a reduction in conformational entropy, new metrics of information and a better understanding of receivers are needed to approach the sequence-conformational ensemble-function relationship of IDRs.

It is interesting to note the proposed role of IDRs in information transfer as linkers, effectors, and sensors that enable complex regulatory behavior in molecular signaling [182,183,184,185,186,187,188]. In this framework, IDRs that connect folded domains can be used as linkers to allosterically propagate information between the sensor and effector domains. IDRs that adopt a structure upon interactions can act as effectors, directly impacting the signaling outcome. IDRs can also be effective sensors, as they can alter the distribution of conformers in their ensembles, both through external signals (e.g., environmental changes, small molecule binding) and internal signals (e.g., PTMs). Finally, IDRs that retain substantial disorder in complexes have been proposed to act as high-capacity information transfer channels, given that their high conformational disorder in the bound-state leads to a large number of interface configurations [182].

Finding correlations between the sequences and conformational and phase-separation propensities of IDRs in a functionally relevant context will continue to be important for understanding disease, as around 20% of disease-associated mutations are found in IDRs [189]. Recently, a proteomics study investigated the impact of disease-associated missense mutations in human IDRs on protein-protein interactions [190]. It was shown that the gain of di-leucine motifs in disordered cytosolic tails of transmembrane proteins can result in protein mistrafficking and might represent a general disease mechanism for the cytosolic IDRs of membrane proteins [190]. In addition to expanding the understanding of disease-associated mutations in SLiMs, it will also be important to investigate the impact of mutations that fall outside of these regions on the conserved molecular features of IDRs.

In the last decades, considerable efforts have been made to derive the sequence-conformational ensemble-function relationship for IDRs. As an increasing number of IDRs are being characterized and patterns in the complicated relationship are becoming clearer, continuous progress will be ensured by nurturing communication and synergy across physical and biological sciences.

## Figures and Tables

**Figure 1 entropy-21-00662-f001:**
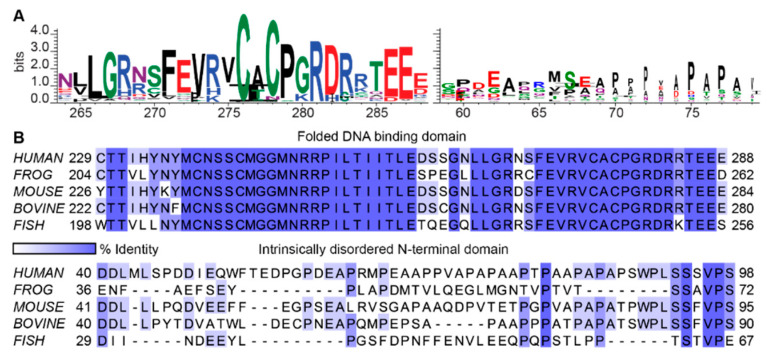
Entropy in alignment reveals information at position according to the positional paradigm. (**A**) *Left*, the ‘sequence logo’ generated for a portion of the DNA binding domain (DBD) of the tumor protein p53 based on the multiple sequence alignment of 69 vertebrate orthologues of p53. *Right*, the ‘sequence logo’ of the intrinsically disordered N-terminal domain (NTD) of p53 based on the same alignment. The numbered positions on the x-axis correspond to the positions in the human sequence. Both logos were generated using WebLogo [34]. (**B**) Multiple sequence alignment of a portion of the folded DNA binding domain (*top*), and intrinsically disordered N-terminal domain (*bottom*) of p53. The alignment is displayed for several vertebrate orthologues. In contrast to folded domains, IDRs display low positional conservation.

**Figure 2 entropy-21-00662-f002:**
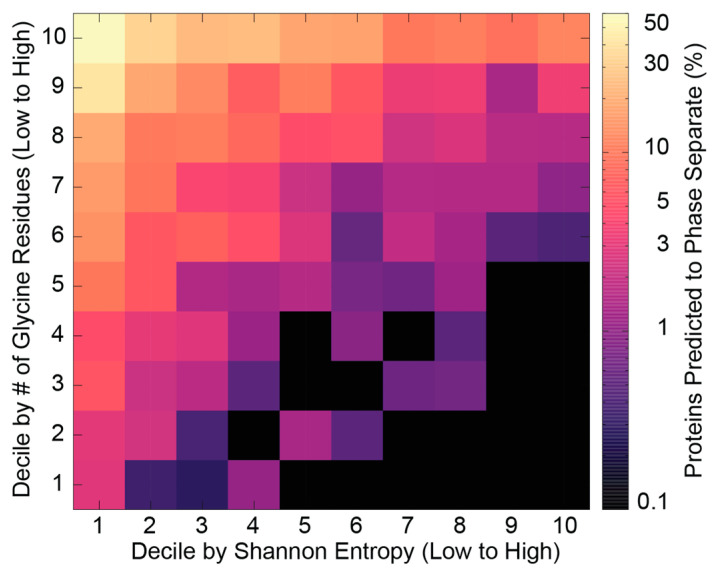
Sequence complexity and glycine content are predictive of phase-separation propensity, shown here for the human proteome across bins ranked by decile position values for glycine content and Shannon entropy of a full sequence. Complexity alone is not sufficient to explain phase-separation behavior; composition differences have additional information even when comparing proteins with similar complexity.

**Figure 3 entropy-21-00662-f003:**
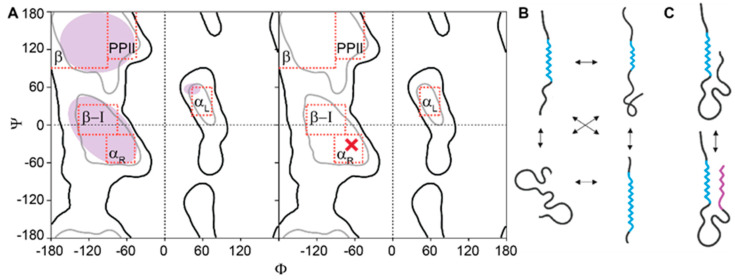
(**A**) The large conformational entropy of IDRs is evident in the broad distribution of the backbone dihedral *Φ*,*Ψ* angles that can be sampled by a single residue in the polypeptide chain (purple, left panel, see references [105,106]). Given the large extent of conformational sampling by every single residue, the total extent of conformational sampling of an entire IDR chain quickly reaches astronomical dimensions as the length of the chain increases. In contrast, a residue in a folded domain in a similar chemical environment, i.e., with identical neighboring amino acids, will typically sample a well-defined set of *Φ*, *Ψ* angles (red cross) that deviates within a very narrow distribution due to thermal fluctuations. Regions of *Φ*, *Ψ* angles that are associated with secondary structural elements are indicated with text on the Ramachandran plot, with favored and allowed areas respectively denoted by the grey and black contours. (**B**,**C**) An illustration of conformational plasticity of an IDR ensemble in the free state based on the works of Schneider et al. [107] and Iešmantavičius et al. [108]. An ensemble of IDRs can feature a range of transient secondary structure (**B**). Modulation of the secondary structural propensities in one part of an IDR can have a stabilizing effect on the secondary structure formation in a distant region of the IDR through transient tertiary contacts (**C**). Transiently formed α-helical segments are indicated in cyan and purple**.** Note that IDRs need not sample any transient secondary or tertiary structure to an appreciable degree in order to be functional (see text).

**Figure 4 entropy-21-00662-f004:**
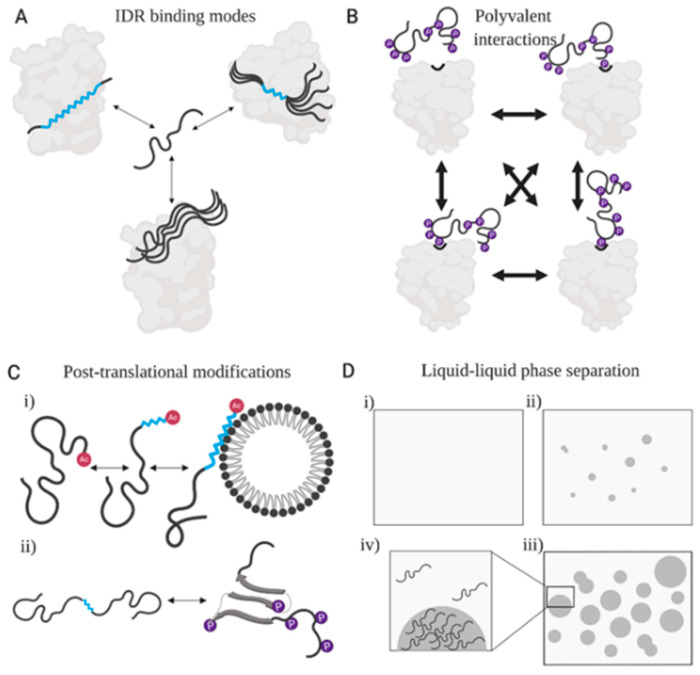
An illustration of functionally relevant processes that impact the configurational and conformational entropy of IDRs. (**A**) IDRs can undergo coupled folding and binding [137] (top left), partial ordering of one part of the chain with high disorder retained in the rest of the chain [131] (top right), or remain highly disordered (bottom middle) in the complex [138]. The parts of the IDR that retain disorder are shown in black and α-helical regions are in cyan. (**B**) PTMs of SLiMs in an IDR can enable polyvalent interactions, whereby multiple sites on the IDR can dynamically exchange with a single binding site of the partner, as illustrated here based on the example of phosphorylated Sic1 in a dynamic complex with Cdc4 (see Mittag et al. [139]). The presence of multiple phosphorylated sites acts to fine-tune the binding affinity by modulating the charge of the IDR. Phosphate groups are shown as purple spheres. (**C**) (**i**) PTMs can induce transient sampling of secondary structure, which can be further reinforced upon functional interactions. The illustrated example is based on the work from Maltsev et al. [140] on α-synuclein. Acetylation of the N-terminus of α-synuclein leads to transient sampling of an α-helical conformation in the first 12 residues that increases the binding affinity for lipids, through enhancing the association rate. The helical conformation is further propagated in the complex of α-synuclein with lipids. (**ii**) PTMs can induce more significant structural transitions, as demonstrated for 4E-BP2 by Bah et al. [133]. Phosphorylation at particular sites leads to formation of a β-sheet, with the extent of phosphorylation impacting the stability of the formed structure. The formation of the β-strands encompasses a sequence motif that, prior to phosphorylation, transiently samples an α-helical conformation and engages in interactions with the binding partner eIF4E. The formation of the β-strand occludes the interaction site on 4E-BP2 thereby weakening the affinity for the binding partner and regulating the interaction to allow downstream functional consequences (i.e., translational initiation by eIF4E). (**D**) Liquid-liquid phase separation manifests as the separation, and the subsequent stable coexistence, of protein-dense (dark grey circles) and protein-dilute (light grey) phases (**ii**,**iii**) from an initially miscible protein solution (**i**). In vitro studies of phase-separation provide the basis for understanding the driving forces of the formation of membraneless organelles in cells [64]. The condensation into the dense phase leads to a decrease in the roto-translational freedom of an IDR (i.e., lower configurational entropy); however, IDRs can still retain high conformational entropy in the condensed state (**iv**) [68,69,70,75]. PTMs can either increase or decrease the propensity of an IDR to phase separate, depending on the overall physicochemical properties of the IDR and the nature of the PTM (see text). The illustration was created with BioRender.com.

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
