# Peer review of "Entropy and Information within Intrinsically Disordered Protein Regions"

_entropy, 2019, doi:10.3390/e21070662_

Round 1

Reviewer 1 Report

Review result is in an attached file.

Author Response

We thank the reviewer for his/her favorable consideration of our perspective and for raising concerns that helped us clarify the content and present the ideas in a clearer way.

Response to Concern1:

We changed the title to “Entropy and information within intrinsically disordered protein regions” to omit this misleading message. The title is chosen to reflect the generality of the presented discussion. We also clarified our discussion with regard to the incomplete loss of entropy, in Sections 3.2 and 3.3.

            Response to Concern2:

            We thank the reviewer for pointing this out. We have now merged those citations, as well as citations in other places with the same issue.

Reviewer 2 Report

Manuscript ID: entropy-529163

Type of manuscript: Perspective

Title: Large entropy does not imply lack of information within intrinsically

disordered protein regions

Authors: Iva Pritišanac, Robert M Vernon, Alan M Moses *, Julie D Forman-Kay

      In this Perspective article, the authors address the entropy content in intrinsically ordered proteins (IDPs). The two basic premises in this paper are: (1) the large positional entropy of intrinsically disordered regions (IDRs) in amino acid sequence space, and (2) the large conformational entropy of IDRs do not necessarily imply a lack of information in IDRs. As a perspective article, I think this manuscript captures some of the relevant experimental and theoretical ideas in this area, and it does a good job summarizing some of the recent activities in the IDP field, particularly as they relate to entropy. So I recommend publication. But I do have some suggestions which I hope the authors will consider. These comments are not meant to be critical. I am just expressing them because I think the readers will be equally curious about the answer.

      At first, when I agreed to review this article, I was very much looking forward to getting a deeper insight into the question that was posed by the paper -- if IDRs have large sequence-space entropy as well as conformational-space entropy, then what kind of "information" do IDRs actually contain? But after going over the paper, I found myself somewhat disappointed. The article sets up the question, but it does not actually provide a definitive answer.

      I think the main issue is that the authors are conflating the different kinds of "information" that are contained in IDRs, and as a result, the article does not achieve in clearly delineating the relationships among them. Clearly, "intrinsically disordered" does not imply lack of "function". Physiological function itself connotes a high degree of information, because out of the many possible amino acid sequences and all the possible functions they could possibly have (or not have), only specific protein sequences have high efficacy as well as high specificity to each physiological function. Out of the thousands of protein sequences that are encoded by the genome, each protein is responsible for a unique function. Therefore, the real question is: What is the proper entropy function that measures biological function? I think the article provides some clues, but it hasn't connected the dots.

      Proteins functions are mediated by molecular interactions, and molecular interactions only operate at certain distance scales, so the structure-function relationship principle is well founded. To be in the functional "on" state, conventional proteins must be correctly pre-folded. But many IDRs can be switched from an "off" state into the "on" state via certain extrinsic factors, whether it is binding to an effector molecule, alignment against other molecular players at microphase boundaries, post-translational modifications, etc. So it is NOT the intrinsic entropy of IDRs in the "off" state that matters, it is actually the entropy CHANGE they undergo when they switched into the "on" state that holds the key. "Intrinsically disordered" means "conformationally diverse", so conformational information is intrinsically low in IDRs. But the more important questions is what is delta H when it switches to the functional state? I think in most cases, there will be a large entropic depression in the conformational ensemble, i.e. applying the condition Y to Eq.(1.4), one would observe a large change in H. And on a higher level in sequence space, I think a similarly large delta H would surface if Y is the condition that the sequence be functional.

Author Response

            We thank the reviewer for their constructive criticism and for pointing out the confusing delivery of our ideas on the topic. We restrained ourselves from making definitive statements about the “entropic depression” upon functional engagements of IDRs, as not all reports in the literature support such a mechanism (please see Section 3.5). Nonetheless, we now made our conclusions with regard to entropy explicit in the Sections 3.2, 3.3, and 3.4. In addition, we added a new Section 3.5 in which we discuss changes in entropy in a functional context, and the need for new entropy functions that can extract information from IDR ensembles. We also extended our discussion in Section 4 (Conclusions and Outlook), to reflect the insight provided by the reviewer.

Reviewer 3 Report

This manuscript is a good survey paper about IDRs of proteins. The limitations of the application of the canonical sequence-structure-function paradigm to IDRs are discussed with respect to two points, i.e., positional information in sequence alignments and conformational entropy in thermodynamics of protein folding. Existing papers are summarized in an intuitive way. This survey can serve as a motivation for developing new methods to study not only IDRs but also structured protein, casting some light on the diversified loop regions with some function such as the complementarity determined regions of immunoglobins and T-cell receptors and on a detailed picture of the denatured state of structured protein.

I have only one comment. I cannot understand what is represented in Figure 4B. This figure is significantly different from Figure 4 in the paper of Mittang et al. [141] the authors cite.

Author Response

We thank the reviewer for his/her favorable consideration of our manuscript.

Response to reviewer's comment:

We thank the reviewer for noting this problem. The presented figure is a simplified version of Figure 4 from Mittag et. al. (PNAS 2008). We now edited the figure to indicate a single phosphate binding site on the receptor, and to include the unbound Sic1 and Cdc4 to the illustrated equilibrium. We kindly ask the reviewer to confirm if this was the issue, or, if this is not the case, to further clarify why the figure is difficult to comprehend.

Round 2

Reviewer 2 Report

The authors have adequately addressed my concerns and I recommend publication.